# Comparative Genomic Analysis of Three *Pseudomonas* Species Isolated from the Eastern Oyster (*Crassostrea virginica*) Tissues, Mantle Fluid, and the Overlying Estuarine Water Column

**DOI:** 10.3390/microorganisms9030490

**Published:** 2021-02-27

**Authors:** Ashish Pathak, Paul Stothard, Ashvini Chauhan

**Affiliations:** 1Environmental Biotechnology Laboratory, School of the Environment, 1515 S. Martin Luther King Jr. Blvd., Suite 305B, FSH Science Research Center, Florida A&M University, Tallahassee, FL 32307, USA; ashish1.pathak@famu.edu; 2Department of Agricultural, Food and Nutritional Science, University of Alberta, Edmonton, AB T6G2P5, Canada; stothard@ualberta.ca

**Keywords:** oysters, comparative genomics, *Pseudomonas*, denitrification

## Abstract

The eastern oysters serve as important keystone species in the United States, especially in the Gulf of Mexico estuarine waters, and at the same time, provide unparalleled economic, ecological, environmental, and cultural services. One ecosystem service that has garnered recent attention is the ability of oysters to sequester impurities and nutrients, such as nitrogen (N), from the estuarine water that feeds them, via their exceptional filtration mechanism coupled with microbially-mediated denitrification processes. It is the oyster-associated microbiomes that essentially provide these myriads of ecological functions, yet not much is known on these microbiota at the genomic scale, especially from warm temperate and tropical water habitats. Among the suite of bacterial genera that appear to interplay with the oyster host species, pseudomonads deserve further assessment because of their immense metabolic and ecological potential. To obtain a comprehensive understanding on this aspect, we previously reported on the isolation and preliminary genomic characterization of three *Pseudomonas* species isolated from minced oyster tissue (*P. alcaligenes* strain OT69); oyster mantle fluid (*P. stutzeri* strain MF28) and the water collected from top of the oyster reef (*P. aeruginosa* strain WC55), respectively. In this comparative genomic analysis study conducted on these three targeted pseudomonads, native to the eastern oyster and its surrounding environment, provided further insights into their unique functional traits, conserved gene pools between the selected pseudomonads, as well as genes that render unique characteristics in context to metabolic traits recruited during their evolutionary history via horizontal gene transfer events as well as phage-mediated incorporation of genes. Moreover, the strains also supported extensively developed resistomes, which suggests that environmental microorganisms native to relatively pristine environments, such as Apalachicola Bay, Florida, have also recruited an arsenal of antibiotic resistant gene determinants, thus posing an emerging public health concern.

## 1. Introduction

Estuaries in the Gulf of Mexico region contribute as much as 69% towards the total harvest of the wild Eastern oysters (*Crassostrea virginica*) in the United States (2016 data, see [1]). Oysters are significant from an economic standpoint as seafood as indicated by a survey in 2019 by the Food and Agriculture Organization of the United Nations (FAO), which valued global wild oyster fishery exports at $148 million USD and aquaculture at $6.8 trillion USD [2,3]. This value continues to increase substantially mainly due to a surge in seafood demands on a global scale [4]. Other than a source of seafood, oysters also serve as keystone species [2,3], by providing a unique habitat for commercially important fish, while at the same time, also performing beneficial ecosystem services such as water filtration and sequestration of excess nitrogen, which is a major cause of coastal and estuarine eutrophication [3,5,6,7]. In fact, it is well known that a single adult oyster can filter as much as 50 gallons of water in 24 h (https://www.fisheries.noaa.gov/national/habitat-conservation/oyster-reef-habitat accessed on 20 January 2021), thus concentrating estuarine microbiota within their nutrient-rich mucosa and digestive organs [8,9,10,11,12]. The enriched suite of oyster microbiota thus likely perform a variety of the above stated beneficial functions, including maintaining functions of the host gastrointestinal tract, providing nutrition, such as vitamins, enzymes, and essential fatty acids, and influencing immune responses and disease resistance. However, limited information exists on several aspects of the oyster microbiome, especially regarding their role in biogeochemical processes, such as maintenance of water quality by recycling dissolved nitrogen. It is known that oysters incorporate nitrogen from the phytoplanktonic biomass they consume into their own tissues but some of the nitrogen is also deposited to the sediment as feces or pseudofeces or excreted into the water column as ammonia [13]. Of major interest is the role that oyster-associated microbial communities likely play in the estuarine N cycling. In fact, denitrification potential of the eastern oyster was recently demonstrated in a seminal study by coupling qPCR and next-generation sequencing approaches [14]. Studies have also suggested symbiotic associations between the oysters and their microbiomes; Colwell and Liston [8] first suggested the existence of a defined commensal flora in Pacific oysters (*Crassostrea gigas*). The findings of Zurel et al. [15] revealed a conserved seasonal association between the Chama-associated oceanospirillales group (CAOG) of bacteria and Pacific oysters which also likely represents a symbiotic association. Trabal et al. [12] reported a symbiotic host–bacteria relationship during different growth phases of two oyster species—*Crassostrea gigas* and *Crassostrea corteziensis*. Such symbionts may assist in the digestion processes, as has been demonstrated in the larvae of *Crassostrea gigas* [16], and may also supply the bivalve host with vitamins and amino acids that serve as growth factors—as shown in the Pacific vesicomyid clam—*Calyptogena magnifica* found at cold seeps [17]. Moreover, certain symbiotic bacteria can even protect their host from pathogens by either producing antimicrobial agents, or by growing in high densities that prevents colonization by other strains [18]. Recently, it was also demonstrated that the oyster extrapallial fluid (EF), which is the site for shell formation, also harbors diverse and abundant microbial communities; moreover, the oyster EF microbiota were found to be enriched for dissimilatory nitrate reduction, nitrogen fixation, nitrification, and sulfite reductase genes and it was proposed that sulfate and nitrate reduction may have a synergistic effect on calcium carbonate precipitation, thus indirectly facilitating in shell formation [19]. Microbial differences within the different oyster tissues have also been demonstrated, such that the stomach and guts of the eastern oysters were shown to be distinct relative to the other organs such as gills, and even the water and sediments that feeds the oyster reefs [9,11,20,21,22]. Specifically, substantial differences between stomach and gut microbiomes of oysters from Lake Caillou in Louisiana were shown such that bacteria belonging to Chloroflexi, Mollicutes, Planctomycetes, and Spartobacteria comprised the major core of the oyster stomach microbiome, whereas Chloroflexi, Firmicutes, α-proteobacteria, and Verrucomicrobia were more abundant in the gut [9]. Therefore, it is clear that the oyster associated microbiota have a profound influence on their hosts and by obtaining a better understanding on the oyster microbiome, perhaps the health and productivity of oysters can be enhanced, which are on the decline. Indeed, several studies have unequivocally shown that the ecological and economical value of oyster habitats continue to deteriorate, especially by eutrophication with excess nitrogen (N) or accidental oil spills, such as the Deepwater Horizon (DWH) oil spill in 2010 [23] (https://news.cornell.edu/stories/2017/01/after-deepwater-horizon-spill-oyster-size-did-not-change; https://cen.acs.org/articles/90/web/2012/11/Deepwater-Horizon-Oil-Spills-Oyster.html; https://www.earthisland.org/journal/index.php/articles/entry/oyster_beds_still_empty_four_years_after_deepwater_horizon_oil_spill/; accessed on 20 January 2021). The continuous dwindling of oyster populations recently resulted in Florida Fish and Wildlife agency to unanimously vote and shut down oyster harvesting in Apalachicola Bay (https://myfwc.com/news/all-news/oyster-commission-1220/ accessed on 20 January 2021), through the end of 2025, dealing a severe blow to this area which historically produced 90% of Florida’s oyster harvest and 10% for the entire United States. Given that the oyster-associated microbiota performs an integral role in maintaining the water quality of the oyster reefs as well as their host organisms, it is necessary to obtain a deeper understanding on the suite of microorganisms that are native to the eastern oysters; an area of research on which not much information currently exists. Towards this end, our ongoing studies using culture-dependent and independent approaches applied to wild oysters have revealed predominant communities to belong to phylum Cyanobacteria (50–75%) in the oyster tissues, gut, and the mantle fluid [24]. Among the bacterial genera within the oyster tissue, *Photobacterium* spp. predominated (50 to 80%); both pathogenic and symbiotic traits are associated with *Photobacterium* spp. [25]. In a more recent survey of cultured diploid vs. triploid eastern oysters from an aquaculture site in the Florida panhandle, conducted over multiple seasons, we found that at the phylum level, proteobacteria predominated in the oyster tissues, followed by Firmicutes, Chloroflexi, Bacteroidetes, and Acidobacteria. Interestingly, at the genus level, between 71% and 77% of the oyster-associated bacterial communities remained taxonomically unresolved, and by inference, may be potentially novel [26]. Among the communities that could be taxonomically resolved, *Lactobacillus* spp. were most abundant, which likely provides critical ecological benefits to oysters, such as disease and stress resistance by production of antibacterial compounds, as well as improving feed utilization and promoting growth in shellfish [27,28]. Overall, it has been shown that when shellfish, including crustaceans, mollusk, and Echinodermata, are treated with specific lactobacilli, it can significantly improve immunity and increase survival rates post pathogen infections, and produce antagonistic effects against *Vibrio* spp. [29]. This area of research needs further assessment, especially on the eastern oyster.

It is interesting to note that in addition to *Lactobacillus* species, there are several other bacterial genera that are used in shellfish aquaculture as probiotics. These include *Enterococcus*, *Bacillus*, *Aeromonas*, *Alteromonas*, *Arthrobacter*, *Bifidobacterium*, *Clostridium*, *Microbacterium*, *Paenibacillus*, *Phaeobacter*, *Pseudoalteromonas*, *Pseudomonas*, *Rhodosporidium*, *Roseobacter*, *Streptomyces*, and *Vibrio* [30]. Towards this end, our previous culture-based analysis of oysters spiked with Gulf crude oil revealed the overrepresentation of *Pseudomonas* species in the isolates that were retrieved [31]. Specifically, 72 strains were isolated, belonging to 10 different genera across the oil-enriched microcosms with 80% of the isolated strains belonging to the *Pseudomonas* genus. This is in line with previous reports that have shown *Pseudomonas* genus to predominate in both raw and retail oysters [32]. It could be that pseudomonads grew rapidly in the presence of high concentrations of nutrients that were provided in the oil-spiked microcosms and outcompeted other slow-growing bacteria that typically have lower Ks (Monod growth coefficient) and low maximum growth rates [33,34]. Furthermore, in our recent study that utilized shallow shotgun sequencing on farmed diploid and triploid oysters collected over several time points over an annual cycle between 2016 and 2017 indicated *Synechococcus*, *Psychrobacter* and *Pseudomonas* as the top three dominant bacterial species [35]. Given the predominance of pseudomonads in the eastern oysters, as demonstrated by our culture-dependent and culture-independent studies, we are tempted to hypothesize that this genus provides a host of services, such as probiotic support, to their host, but this needs to be tested. Regardless, our findings are consistent with several other studies that showed bacteria from the Pseudomonadaceae family to rapidly respond in the biodegradation of oil hydrocarbons [36,37,38], most likely due to the presence of a suite of hydrocarbonoclastic genes. However, a detailed genome-wide comparative analysis on the eastern oyster-associated *Pseudomonas* species, especially from the Gulf of Mexico ecosystem is lacking. To advance this area of research, this comparative genomic analysis was performed on three previously isolated *Pseudomonas* strains, which include *P. alcaligenes* strain OT69 (isolated from minced oyster tissue); *P. stutzeri* strain MF28 (isolated from oyster mantle fluid), and *P. aeruginosa* strain WC55 (isolated from the water collected from top of the oyster reef) [31,39]. Comparative genomic analysis on these targeted microbial strains native to the eastern oyster and its surrounding environment revealed unique functional traits and conserved gene pools between species as well as those genes that render unique characteristics in context to their evolutionary history and adaptive traits. Furthermore, this study significantly advances our understanding on the repertoire of metabolic traits possessed by the oyster-associated pseudomonads, on which limited information is currently available.

## 2. Experimental Section

### 2.1. Isolation of Oyster-Associated Pseudomonas Strains

Several oyster-associated microorganisms were isolated from Dry bar (29°40.474 N, 85°3.497 W), which was one of the most productive oyster harvesting site in Apalachicola Bay, Florida [31,39]. Because *Pseudomonas* species were found to predominate in lab-controlled microcosms spiked with Gulf crude oil [31], we obtained draft genomes of 3 strains which were isolated from oiled microcosms, as shown before [31,39]. Specifically, the sources of these isolates were the oyster tissue (*Pseudomonas alcaligenes* strain OT69); oyster mantle fluid (*Pseudomonas stutzeri* strain MF28); and water samples collected from top of the oyster reef (*Pseudomonas aeruginosa* strain WC55), respectively. Draft genome sequences of these 3 strains were previously reported [39], on which further comparative genomic analysis is performed and reported herein.

### 2.2. Nucleotide Sequence Accession Number

The draft genome sequences of the strains obtained in this study have been deposited as whole-genome shotgun projects in GenBank under the accession numbers ATAQ00000000 (*Pseudomonas aeruginosa* WC55), ATCP00000000 (*Pseudomonas alcaligenes* OT69), ATAR00000000 (*Pseudomonas stutzeri* MF28).

### 2.3. Genome-Wide Comparison of Pseudomonas Strains

A circular genomic map of the three oyster-associated *Pseudomonas* strains was constructed using the CGView comparison tool [40], relative to *Pseudomonas putida* KT2440 as the reference strain. Phylogenomic analysis to identify nearest neighbors was conducted using the Onecodex workflow [41], which is based on the identification of short sequences (17–31 bp), that are unique to a specific taxon within the inputted sequence reads. Based on the collection of k-mers found in a given read, the whole genome sequence is then assigned to a specific taxon. Phylogenomic analysis was run using the default value of k = 31 for the taxonomic assessment of *Pseudomonas* strains using the Onecodex database.

Taxonomic affiliations of isolated *Pseudomonas* spp. were inferred at the genomic level using the Type (Strain) Genome Server (TYGS), a free bioinformatics platform available under https://tygs.dsmz.deIMG (accessed on 2 December 2020), for a whole genome-based taxonomic analysis [42]. For the phylogenomic inference, pairwise comparisons among the set of the three *Pseudomonas* genomes were conducted using the Genome BLAST Distance Phylogeny method (GBDP) that resulted in accurate intergenomic distances inferred using the embedded algorithm "trimming" and distance formula d5; 100 distance replicates were calculated for each iteration. Digital DDH values and confidence intervals were calculated using the recommended settings of the previously reported GGDC web tool 2.1 [43]. The intergenomic distances obtained after this analysis were used to infer a balanced minimum evolution tree with branch support via FASTME 2.1.4 including SPR postprocessing [44]. Branch support was inferred from 100 pseudo-bootstrap replicates for each analysis; resulting trees were rooted at the midpoint and visualized with PhyD3 [45].

The genome annotations and prediction analysis were conducted using Rapid Annotations using Subsystems Technology-RAST [46], Prokaryotic Genomes Automatic Annotation Pipeline (PGAAP), version 2.0 [47], the Pathosystems Resource Integration Center (PATRIC, version 3.6.5) [48], or Integrated Microbial Genomes (IMG) system [49]. Specialty genes, such as for virulence, drug targets, and antimicrobial resistance were identified using PATRIC and the Comprehensive Antibiotic Resistance Database (CARD) [50]. Average nucleotide identity (ANI), and average amino acid identity (AAI) was also obtained via EDGAR [51] and EzBioCloud pipelines [52]. ANI is a measure of the genomic resemblance of two different bacterial species and typically, ANI values between genomes of the same species are 95% or beyond [53].

Genomic and phylogenomic comparisons were further conducted using the EDGAR pipeline [51]. After EDGAR analysis, the Newick phylogenomic tree file was downloaded and a nucleotide-based tree was constructed using MEGAX [48,54]. IslandViewer was used to locate chromosomal deviations in GC content, also known as genomic islands (GEIs) [55]. Remnants of phage DNA, also called prophage, present in the whole genome sequence of isolated Pseudomonads were identified using PHASTER [56]. Evolutionary relatedness of the three pseudomonas strains were evaluated using Mauve [57], which performs multiple genome alignments to identify genome-wide rearrangements and inversions resulting from evolutionary events. Genomic recombination events leaves orthologous genomic regions of a bacterial strain as reordered or inverted genomic regions relative to related genomes, which are clearly identified using the Mauve analysis; conserved genomic segments that appear to be internally free from rearrangements are shown as Locally Collinear Blocks (LCBs).

## 3. Results and Discussion

### 3.1. Genomic Features of the Isolated Pseudomonas Strains

Three *Pseudomonas* species were isolated and characterized for their genomic traits, as shown before [39]. Briefly, these three strains were isolated from oyster tissue (*Pseudomonas alcaligenes* strain OT69); oyster mantle fluid (*Pseudomonas stutzeri* strain MF28); and estuarine water samples (*Pseudomonas aeruginosa* strain WC55), respectively. Specifically, draft genome sequencing analysis was performed on these strains previously and published as a genome announcement [39]; however, a detailed comparative genomic analysis was not attempted, which is the overarching objective of this study. Further analysis of these 3 strains revealed genomic sizes of 7 Mp (strain OT69), 4.9 MB (strain MF28), and 6.8 MB (strain WC55), respectively. N50 values ranged from 92,012 to 128,670 with GC contents between 62.29 to 66.14. Total number of putative genes were 6543 (strain OT69), 4630 (strain MF28), and 6650 (strain WC55). Of significant interest were the relatively higher number of genes for xenobiotic degradation, ranging from 813 to 976, which will be detailed later in this report. A comparative circular genomic map of the 3 isolated *Pseudomonas* strains relative to *Pseudomonas putida* KT2440 is shown in Figure 1.

Interestingly, whole genome based taxonomic analysis of isolated strains using the TYGS workflow suggested that strains isolated from the oyster tissue and mantle fluid are potentially new species (Table 1). Specifically, pair-wise comparison of strain *Pseudomonas alcaligenes* strain OT69 isolated from the oyster tissue, against the type-strain genomes yielded a digital DDH (d4) value of 29% with *Pseudomonas otitidis* DSM 17224 as the top hit; according to the TYGS workflow; d4 values reflect the sum of all identities found in high scoring pairs (HSPs) divided by overall HSP length and d4 is independent of genome length, thus, is more robust against the use of incomplete draft genomes as compared to other dDDH formulae used by TYGS analysis. The closest *Pseudomonas alcaligenes* to strain OT69 was found to be *Pseudomonas alcaligenes* NBRC 14159 with a d4 value of 24.4%. The overall conclusion of the TYGS analysis indicated strain OT69 to be a potentially new species (Table 1). Similarly, strain MF28, isolated form the oyster mantle fluid also appears to be a potentially new specie. Specifically, strain MF28 analysis yielded a digital d4 value of 24.9% with *Pseudomonas xanthomarina* DSM 18231 as the top hit. The closest *Pseudomonas stutzeri* to strain MF28 was found to be *Pseudomonas stutzeri* ATCC 17588 with a d4 value of 22.8% (Table 1). Conversely, for strain WC55, isolated from the estuarine water of the oyster bar, the d4 value of 94.7% was obtained with the closest match with *Pseudomonas aeruginosa* DSM 50071 (Table 1), and hence is a known specie of the *P. aeruginosa* group. Phylogenomic trees obtained from the TYGS analysis are shown in Figure 2A–C.

Hierarchical cluster analysis based on the presence of COGs (Clusters of Orthologous Genes), via the img/er workflow, revealed a better taxonomic affiliation for each of the three isolates such that *Pseudomonas alcaligenes* strain OT69 was found closest to *P. alcaligenes* strains MRY13_0052 and S-00099; *Pseudomonas stutzeri* strain MF28 was closest to strains SDM_LAC and NP_8Ht and *Pseudomonas aeruginosa* strain WC55 was closest to strains PA7790 and MH27, respectively, as shown in Appendix A. It must be noted that COGs represent an ortholog or direct evolutionary counterpart among bacterial genomes as they evolve over time and hence serve as a better approach for phylogenetic analysis of the pseudomonads isolated for this study. Overall, the TYGS based analysis suggests that the oysters remain a largely underexplored niche that likely harbors novel microbiota, as even the cultured *Pseudomonas* representatives from the oyster tissue and mantle fluid appear to be novel, unlike the one isolated from the water column. This is in line with our recent metagenomic studies which also revealed that between 71% and 77% of the oyster-associated bacterial communities remained taxonomically unresolved, and by inference, may be potentially novel [26], and by inference, most likely novel. Notably, in our most recent study that utilized shallow shotgun sequencing on diploid and triploid oysters collected over several time points over a year between 2016 and 2017, indicated *Scenecococcus*, *Psycrhobacter*, and *Pseudomonas* as the top three dominant bacterial species [35]. Therefore, it appears that *Pseudomonas* spp., may represent one bacterial group that is intimately associated with the oyster tissues, as shown by our multi-pronged culture-dependent and culture-independent studies, and thus performing a comparative genomic analysis on this group may reveal further insights on the microbe-oyster dynamics, especially as it relates to the pseudomonads. This is especially relevant because several other studies have also reported that the most commonly retrieved bacteria from the tissues and fluids of bivales, including oysters, clams, and mussels are strikingly similar; mainly belonging to *Pseudomonas* as well as *Vibrio*, *Acinetobacter*, *Aeromonas*, and members of the Flavobacteria/Cytophaga/Bacteroides [8,18,58].

Gene prediction and annotation of the isolated strains was then performed using several different workflows, such as RAST and PATRIC, respectively. Further analysis of *Pseudomonas alcaligenes* OT69 using the RAST-based functional gene subsystem clustering revealed the presence of a total of 1870 subsystems represented by 36% of the strain’s genome. The top five subsystems belonged to carbohydrate metabolism (259); amino acids and derivatives (464); cofactors, vitamins, prosthetic groups, and pigments (137); protein metabolism (203); and fatty acids, lipids, and isoprenoids (115). Additionally, functions related with membrane transport (141); stress response (133); resistance to antibiotics and toxic compounds (124); as well as metabolism of aromatic compounds (64). RAST analysis of strain *Pseudomonas stutzeri* MF28 revealed the presence of a total of 2250 subsystems represented by 49% of the strain’s genome. The top five subsystems belonged to carbohydrate metabolism (398); amino acids and derivatives (455); cofactors, vitamins, prosthetic groups, and pigments (243); protein metabolism (213); and fatty acids, lipids, and isoprenoids (192). Additionally, functions related with membrane transport (182); stress response (170); resistance to antibiotics and toxic compounds (90); as well as metabolism of aromatic compounds (77). Similarly, *Pseudomonas aeruginosa* WC55 consisted of a total of 3019 subsystems represented by 48% of the strain’s genome. The top 5 subsystems belonged to carbohydrate metabolism (442); amino acids and derivatives (687); cofactors, vitamins, prosthetic groups, and pigments (365); protein metabolism (264); and fatty acids, lipids, and isoprenoids (212). Additionally, functions related with membrane transport (240); stress response (189); resistance to antibiotics and toxic compounds (161); as well as metabolism of aromatic compounds (153). It is however worth noticing that only 36% of the genes in strain OT69 were assignable to subsystems using RAST with 64% not annotated to any subsystem; similarly 51% and 52% genes in strains MF28 and WC55 could not be annotated with any existing subsystems in RAST. This suggests that we are unable to obtain an accurate prediction of more than half of these strain’s genomic makeup, which is likely arising because these strains are distantly related any known reference genomes, as shown by the TYGS analysis indicating strains OT69 and MF28 to be potentially novel. As more strains are sequenced, annotated, and made publicly available, this situation may change, and we may accrue more information on the genomic organization and functions of the oyster-associated pseudomonads.

When PATRIC-based annotation of strain *Pseudomonas alcaligenes* OT69 was performed, the presence of 2048 hypothetical proteins and 4317 proteins with functional assignments was found, which included 1229 proteins with Enzyme Commission (EC) numbers, 1164 with Gene Ontology (GO) assignments, and 932 proteins mapped to KEGG pathways. PATRIC annotation includes two types of protein families, and the genome of strain *Pseudomonas alcaligenes* OT69 possessed 6352 proteins each belonging to the genus-specific protein families (PLFams) and the cross-genus protein families (PGFams), respectively. Similar analysis of *Pseudomonas stutzeri* MF28 revealed the presence of 1257 hypothetical proteins and 3320 proteins with functional assignments, which included 1094 proteins with Enzyme Commission (EC) numbers, 1028 with Gene Ontology (GO) assignments, and 831 proteins mapped to KEGG pathways. Strain OT69 possessed 4576 proteins each belonging to the genus-specific protein families (PLFams) and the cross-genus protein families (PGFams), respectively. Finally, the PATRIC-based annotation of *Pseudomonas aeruginosa* WC55 revealed the presence of 1460 hypothetical proteins and 4965 proteins with functional assignments, which included 1306 proteins with Enzyme Commission (EC) numbers, 1206 with Gene Ontology (GO) assignments, and 979 proteins mapped to KEGG pathways. Strain OT 69 possessed 6425 proteins each belonging to the genus-specific protein families (PLFams) and the cross-genus protein families (PGFams), respectively. The presence of 58, 28, and 92 pseudogenes, which are nonfunctional segments of DNA, were also identified in strains OT69, MF28, and WC55, respectively. Further genome scale analysis revealed a significant potential for xenobiotic degradation in the three pseudomonads, which was then further studied.

### 3.2. Xenobiotic Degradation and Speciality Genes in Isolated Strains

A preliminary genomic analysis on the three isolated pseudomonads previously revealed consistently higher abundances of putative genes for xenobiotic degradation and metabolism; some common genes across the isolated strains included those involved in the degradation of toluene, xylene, and PAHs, as well as those for the degradation of chloroalkane, chloroalkene, and nitrotoluene, respectively [39]. Some questions remained unanswered in the previous report, such as other genomic mechanisms by which these strains resist pollutants and engage in biodegradative functions, as well as how their biodegradative genes have evolved relative to other pseudomonads. Given that *Pseudomonas* spp., are well documented to possess a high metabolic activity and potential for hydrocarbon degradation, and can resist a variety of contaminants in diverse environmental habitats [59,60,61], we queried the genomes of the isolated strains in RAST; the total genes for metabolism of aromatic compounds (in parenthesis) are as follows: *Pseudomonas alcaligenes* OT69 (64), *Pseudomonas stutzeri* MF28 (77) and *Pseudomonas aeruginosa* WC55 (153), respectively. We previously reported that *P. aeruginosa* strain WC55 showed the highest utilization of crude oil in lab-controlled microcosm studies, relative to the other two strains [31], which is consistent with the highest number of xenobiotic genes identified in strain WC55. It is likely that the estuarine water microcosms, which was the isolation source of strain WC55, is naturally primed with crude oil seeps, which are known to continuously emit hydrocarbons into the overlaying water column as bubble plumes which are often coated with a thin layer of oil [62,63,64,65]. It is in fact estimated that over 200 active seeps are present in the Gulf of Mexico which emit oil to the Gulf waters at a high rate of 0.4–1.1 × 108 L/year [66]. Therefore, it is to be expected that this “natural” oil primes the environment continuously and selectively enriches the native microorganisms to recruit genomic ability for crude oil metabolism. This seems to be a plausible explanation for the water column microbiota to recruit and sustain a plethora of hydrocarbon degrading genes relative to the oyster-associated microbiota, which are likely protected by the internal habitat of the bivalve shell, and hence genes for xenobiotic degradation were highest in the strain isolated from the water column (strain WC55) relative to strains isolated from the oyster tissue (OT69) and mantle fluid (MF28), respectively.

Outer membrane transport is another major mechanism by which bacteria efflux toxic compounds and may be the first step in bacterially-mediated biodegradation of aromatic compounds [67]. Upon further genomic analysis of these strains, a plethora of membrane transport functions were identified. Specifically, 141, 182, and 240 genes were identified for membrane transport in *Pseudomonas alcaligenes* OT69, *Pseudomonas stutzeri* MF28, and *Pseudomonas aeruginosa* WC55, respectively. ABC transporters were the main category of transport proteins, as well as protein secretion systems, uni-Sym and antiporters, TRAP transporters, Ton and Tol transport systems and other transport systems. Another mechanism by which motile bacteria perform taxis either towards or away from toxic compounds is chemotaxis [68], which increases both, the bioavailability and biodegradative efficacy of aromatic pollutants [69]. Genomic analysis of chemotaxis functions revealed the presence of 85, 157, and 115 motility and chemotaxis genes in strains OT69, MF28, and WC55, respectively.

Furthermore, a number of specialty genes were also identified in the isolates that includes virulence factors, drug targets and antibiotic resistance. Specifically, strains OT69 harbored 49 genes for virulence factors, 28 genes for drug targets, and 81 genes for antibiotic resistance, respectively. Similarly, strain MF28 possessed 28, 25, and 47 genes and strain WC55 possessed 223, 63, and 105 genes for the above stated functions of virulence, drug targets and antibiotic resistance. Further, noteworthy, were the presence of functional traits related to stress response in all the three pseudomonads. Specifically, strain OT69 possessed 133 genes related to stress functions, strain MF28 harbored 170 genes and strain WC55 contained 189 stress related genes, respectively. Most of the stress related genes were related to oxidative stress, followed by osmotic stress, heat and cold shock, and several others not specific to a particular category.

Another feature that is noteworthy was the presence of nitrogen metabolism genes in the isolated strains. As stated earlier in this work, oysters have the ability to filter as much as 50 gallons of water/day, thus cleaning their surrounding water, removing organic matter that can cause low-oxygen “dead zones”, and assisting in sequestration of nutrients like nitrogen (N) and phosphorus (P) from terrestrial runoff thus improving water quality and mitigating algal blooms and fish kills. Therefore, genomic studies can obtain a better assessment of the oyster-borne microbiota and their ecosystem services, especially removal of nitrogen via denitrification processes, which consists of four reaction steps in which nitrate is reduced to dinitrogen gas. The reduction of nitrite (NO2−) to nitric oxide is catalyzed by two different types of nitrite reductases (Nir), either a cytochrome cd1 encoded by nirS or a Cu-containing enzyme encoded by nirK. The reduction of nitrous oxide is the last step in the denitrification pathway and is catalyzed by nitrous oxide reductase encoded by the nosZ gene. The three isolated *Pseudomonas* species were queried for the presence of denitrification genes and it was found that strain OT69 possessed the nitric-oxide reductase subunit B (EC 1.7.99.7; PATRIC ID fig|1333854.3.peg.464), with the closest similarity with *P. denitrificanse* ATCC 13867 (Appendix A). Strain MF28 possessed nitric-oxide reductase subunit B (PATRIC ID fig|1333856.3.peg.3050) (closest similarity with *P. chloritidismutans* AW-1) (Appendix A) as well as the subunit C (PATRIC ID fig|1333856.3.peg.3051) (closest similarity with *P. stutzeri* KF716) (Appendix A), along with a nitrite reductase (EC 1.7.2.1; PATRIC ID fig|1333856.3.peg.4134) (closest similarity with *P. stutzeri* YC-YH1) (Appendix A). Similarly, strain WC55 was found to harbor nitric-oxide reductase subunit B (PATRIC ID fig|1333855.3.peg.904) (Appendix A), subunit C (PATRIC ID fig|1333855.3.peg.905) (Appendix A), and a nitrite reductase (PATRIC ID fig|1333855.3.peg.909) (Appendix A), respectively. These denitrification genes in strain WC55 affiliated closest to *P. aeruginosa* PAO1. It appears that these isolated strains likely possess some degree of nitrogen cycling traits but lack a complete denitrification pathway, which would contain either a nirK or nirS gene along with the nosZ gene.

### 3.3. Comparative Genomic Analysis on the Isolated Strains

An EDGAR based analysis was performed to assess the presence of commonly shared genes as well as those that are unique to each of the three isolated pseudomonads, as shown in Figure 3. When *Pseudomonas putida* KT2440 was used as the reference strain, it was found that the four pseudomonads shared a conserved core genome comprising of 1133 coding DNA sequences (CDSs). *Pseudomonas alcaligenes* OT69 contained 3049 distinct singletons; the largest number of singletons amongst the four strains. *Pseudomonas stutzeri* MF28 contained 1926 unique genes and *Pseudomonas aeruginosa* WC55 contained 2536 unique genes, respectively. Interestingly, when this was viewed in degree three, which is how the genes are shared by every three strains; only 160 genes were found common between strains OT69, MF28 and WC55 (Figure 3), indicating that each of these pseudomonads are uniquely structured in context of their genomic structure. The largest portions of these shared genes between strains OT69, MF28, and WC55 were associated with functions related to biodegradation, efflux, membrane transport, chemotaxis—all of which likely collectively facilitate several ecosystem services to the oyster reef habitat. As stated in the above section, genes for nitrogen cycling were also identified by comparative genomics using EDGAR. Specifically, the copper-containing nitrite reductase (EC 1.7.2.1), nirK gene was identified as part of the unique genes of strain OT69 along with several nitrate/nitrite transporter genes within the genomes of all three strains.

Another interesting genomic trait within bacterial genomes is the presence of genomic islands (GEIs), which are typically acquired via horizontal gene transfer (HGT) mechanisms. Genes encoded on genomic islands can render additional adaptive traits and genomic plasticity driven by exposure to environmental factors, which may thus facilitate evolutionary survival [70,71,72]. Overall, GEIs are classified into four major categories: pathogenicity islands (PAIs) that code for virulence genes; metabolic islands (MIs), possessing genes for secondary metabolites biosynthesis; resistance islands (RIs), which usually code for antibiotics resistance; and symbiotic islands (SIs), which are those genes that code for symbiotic associations with other host species. In the current study we were able to identify several genomic islands integrated into the genomes of the three isolates predicted via the available three methods in the IslandViewer pipeline. Specifically, strain OT69 possessed 50 GEIs when compared against the complete genome of *P. alcaligenes* strain NEB 585 as the reference strain (Figure 4A); these GEIs ranged from 61,670 bp to 4132 bp in size (Figure 4A). Similarly strain MF28, when compared against *P. stutzeri* strain DSM10701possessed 40 GEIs ranging from 64,189 bp to 4255 bp in size (Figure 4B). Strain WC55 contained 21 GEIs ranging from 53,475 to 4505 bp in size (Figure 4C), relative to the genome of *Pseudomonas aeruginosa* PAO1. Noteworthy is the presence of several genes found on GEIs related to the function of pollutant biodegradation, especially the monooxygenase and dioxygenase enzymes. It is likely that these strains were exposed to crude oil in their natural environments via natural oil seep activity and they recruited oxygenase genes as a survival mechanism because these genes facilitate biodegradation of oil components such as polycyclic aromatic hydrocarbons (PAHs), BTEX (benzene, Toluene, Ethylene and Xylene), and other aromatic compounds. Of further interest was the presence of genes related to the function of transport, such as the ATP-binding cassette (ABC) and the major facilitator superfamily (MFS) proteins, chromate, arsenic, zinc, nickel, mercury, and the Resistance-nodulation-division (RND) transporters, as well as chemotaxis protein CheY; these likely facilitate survival of the strains in a hydrocarbon polluted environment, as is well known for other microorganisms.

From an evolutionary standpoint, one lateral DNA transfer mechanism into environmental microbiota occurs via viral-mediated attack(s), which can leave remnants of viral genomic materials within the host bacterial genomes [73,74]. Upon integration of the bacteriophage genome into the bacterial host cell, the viral genome is referred to as a prophage, which can impart additional functional traits to the recipient bacteria, such as virulence factors, enzymes, toxin biosynthesis genes, regulatory factors, and these phage mediated transferred DNA can range from 1 kb to more than 100 kb in size [75]. Analysis of phage remnants in the three *Pseudomonas* strains revealed the presence of several prophage regions, as shown in Figure 5A–C. Specifically, strain OT69 was found to harbor 6 prophage regions, of which four regions were intact and two regions were incomplete (Figure 5A). The prophage remnants sizes were of 53.5 kb, 46.9 kb, 45.4 kb, 36.8 kb, 23.6 kb, and 18.5 kb, respectively. Conversely, strain MF28 was found to sustain only three incomplete prophage regions of 9.7 kb, 9.4 kb, and 8.1 kb in size (Figure 5B). Similarly, this prophage analysis revealed the presence of five prophage regions in strain WC55 with one intact, three incomplete and one was questionable (Figure 5C); these regions were of 42.7 kb, 42.4 kb, 22.8 kb, 19.9 kb, and 17.7 kb in size. These findings were also confirmed by RAST based annotation in which strain OT69 was found to harbor 29 phage and prophage-related genes; similar to strain WC55, which contained 24 phage-associated gene fragments. Conversely, strain MF28 contained only one integron gene fragment. Among the phage-associated genes in strains OT69 and WC55, the predominant ones were for phage packaging machinery, phage tail proteins and phage tail fiber/capsid proteins. The higher number of intact prophages in strain OT69 isolated from the oyster tissues suggests the possibility of higher phage-related attacks to the existing microbiota relative to mantle fluid and water ecosystems and by inference, it can be hypothesized that the oyster tissues represent a previously underexplored “hotspot” of microbial communities which have likely evolved due to host–microbe interactions, including horizontal gene transfer mechanisms, and likely have the potential for producing hitherto unknown biotechnological products, such as antimicrobials and biosynthetic compounds.

Towards this end, when the antiSMASH workflow was used to analyze the genomes for secondary metabolite biosynthesis gene clusters, we found six DNA regions with one region showing 100% similarity to the polyketide synthetic gene cluster of *Pseudomonas fluorescens* (Appendix A). Similarly, strain MF28 possessed nine genomic regions, one of which was 100% similar to the terpene synthesis cluster from the Enterobacteriaceae bacterium DC413 (Appendix A). Interestingly, strain WC55 possessed 15 gene clusters for secondary metabolites, which was the highest amongst these strains (Appendix A); with 100% similarity to the pyochelin biosynthetic gene cluster from *Pseudomonas aeruginosa* PAO1. Overall, these analyses show that these isolated strains should be further screened for antimicrobials. Evaluation of antimicrobial resistance genes, or the resistomes in the 3 isolates using the CARD pipeline (Appendix A), revealed the presence of 58 hits to drug classes in strain WC55; the highest amongst the 3 isolates (Appendix A). Noteworthy were 34 hits against tetracycline, followed by 32 against fluroquinolone. Similarly, strains OT69 and MF28 each contained five gene homologues for fluroquinolone as the main drug against which the strains are likely resistant. These findings are in line with the annotations obtained using PATRIC stated in Section 3.2, which also showed the presence of several specialty genes in these strains, such as for virulence and drug targets. The presence of higher BGCs and resistome in strain WC55 is likely because it is native to the estuarine water, which are directly exposed to environmental contaminants, which also explains higher number of xenobiotic genes in this strain, relative to the oyster tissue and mantle fluid, from which strains OT69 and MF28 were isolated.

Further genomic comparisons between the three isolated Pseudomonads were performed using MAUVE, a workflow designed to construct multiple genome alignments in context to the presence of large-scale evolutionary events such as rearrangements and inversions, thus providing visualizations of evolutionary dynamics between bacterial strains being compared. Mauve analyses showed the presence of several crisscrossing locally collinear blocks (LCBs), suggesting the complicated rearrangement profiles of each strain, as shown in Figure 6. Regions that are inverted between each strain are shifted below a genome’s center axis; further visual inspection of the rearrangement pattern for each strain revealed several instances of local overlapping inversions characteristic of symmetric inversion about the terminus, distinguishable as a “fan” pattern of crossing lines. Overall, this analysis suggests a complex exchange pattern of genomic segments that likely occurred throughout the evolutionary history of the three isolated pseudomonads, likely facilitated by phage attack and reshuffling of incorporated genomic materials within each strain.

Overall, this comparative genomics study revealed a suite of genomic traits within three *Pseudomonas* species isolated from the homogenized oyster tissues, mantle fluid and the estuarine water that feeds the oysters. The studied strains likely recruited their genomic traits during their evolutionary history and exposure to environmental drivers, horizontal gene transfer events as well as phage-mediated incorporation of genes. The genomic traits within the three pseudomonads potentially provide a variety of ecologically and environmentally beneficial services, such as nitrogen transformation and biodegradation of xenobiotic compounds. Moreover, the strains also supported an extensively developed resistome, which suggests that environmental microorganisms native to relatively pristine environments, such as the Apalachicola Bay, Florida, have also recruited an arsenal of antibiotic resistant gene determinants, with the potential to exacerbate and develop into highly potent multi-drug resistant “superbugs,”, with threats to public health as well as causing animal diseases. Therefore, mitigation of antimicrobial resistance should certainly be on the agenda for the immediate future with concerted efforts between scientists, governmental organizations, and policy makers for better environmental stewardship on a global scale.

## Figures and Tables

**Figure 1 microorganisms-09-00490-f001:**
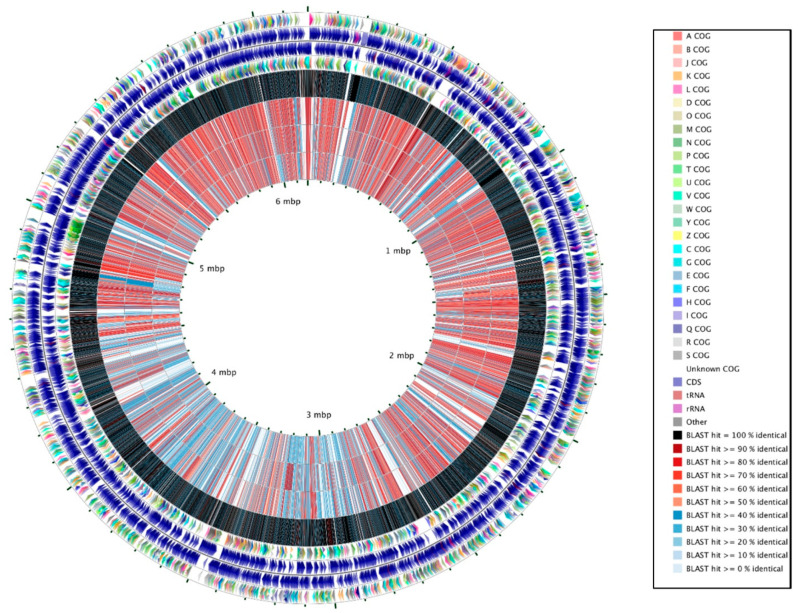
Comparison of three *Pseudomonas* spp., isolated from oyster tissue (strain OT69), water column (strain WC55) and mantle fluid (strain MF28) relative to *Pseudomonas putida* KT2440 as the reference strain, using the CGView Comparison Tool [40]. The outermost two rings depict Clusters of Orthologous Gene (COG) functional categories (first ring) and genes (second ring) on the forward strand of *Pseudomonas putida* KT2440. The third and fourth rings depict COGs and genes, respectively, on the reverse strand. The remaining rings display the results of blastp comparisons between the *Pseudomonas putida* KT2440 (CDS; coding DNA sequences) translations and the CDS translations from *Pseudomonas putida* KT2440, *Pseudomonas alcaligenes* OT69, *Pseudomonas aeruginosa* WC55, and *Pseudomonas stutzeri* MF28. For the BLAST comparisons an E-value cutoff of 0.1 was used. The regions of the reference genome yielding blast hits are indicated within each BLAST results ring, and each region is colored according to the percent identity of the hit.

**Figure 2 microorganisms-09-00490-f002:**
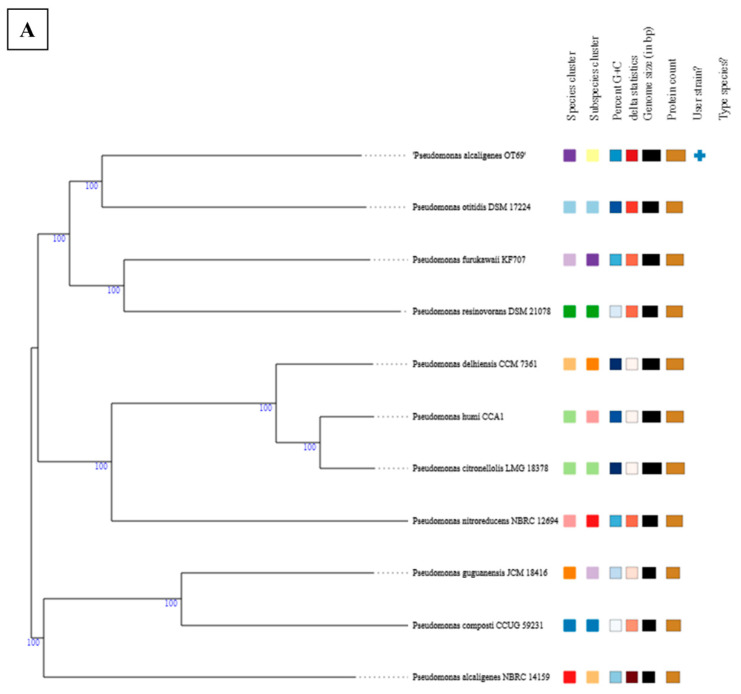
Phylogenomic trees obtained from the TYGS pair-wise comparison of *Pseudomonas alcaligenes* strain OT69 (**A**), *Pseudomonas stutzeri* strain MF28 (**B**) and *Pseudomonas aeruginosa* strain WC55 (**C**), respectively. The + symbol indicates user strain, which represent the isolates obtained in this study and used for this analysis.

**Figure 3 microorganisms-09-00490-f003:**
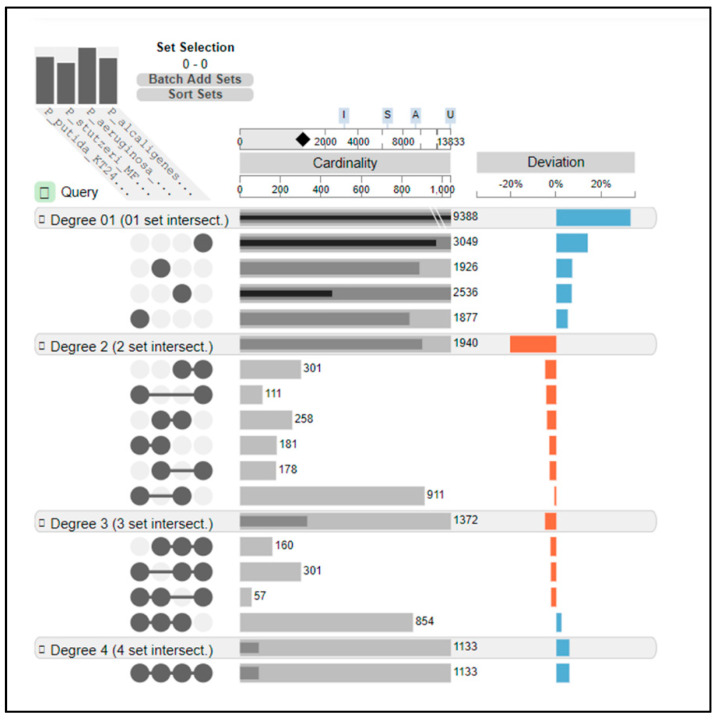
Unique and shared gene profiles of *Pseudomonas alcaligenes* OT69, *Pseudomonas stutzeri* MF28, and *Pseudomonas aeruginosa* WC55 when *Pseudomonas putida* KT2440 was used as the reference strain. The EDGAR workflow platform was used here based on the orthology analysis. These results are visualized and shown in four degrees; degree one represents the distinct singletons harbored by each single strain; degree two represents the genes shared by every two strains; degree three represents the genes shared by every three strains; degree four is the number of core genes shared by all the four strains.

**Figure 4 microorganisms-09-00490-f004:**
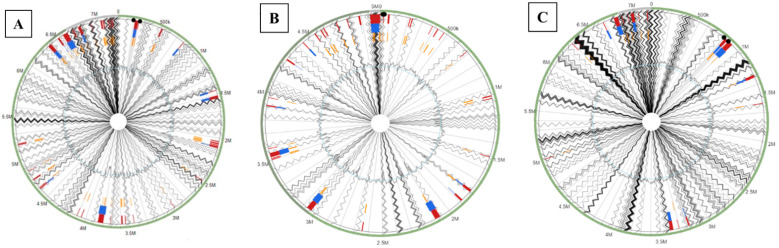
Genomic islands identified within the whole genome sequence of (**A**), *Pseudomonas alcaligenes* OT69; (**B**), *Pseudomonas stutzeri* MF28 and (**C**), *Pseudomonas aeruginosa* WC55. Reference genomes used in this analysis were as follows: *Pseudomonas alcaligenes* strain NEB 585 for strain OT69, *Pseudomonas stutzeri* DSM 10701 for strain MF28 and Pseudomonas aeruginosa PAO1 for strain WC55. The outer black circle represents the scale line in Mbps, and genomic islands (GEIs), obtained from each of the following methods are shown in color: SIGI-HMM (orange), IslandPath-DIMOB (blue), and integrated detection (red), respectively.

**Figure 5 microorganisms-09-00490-f005:**
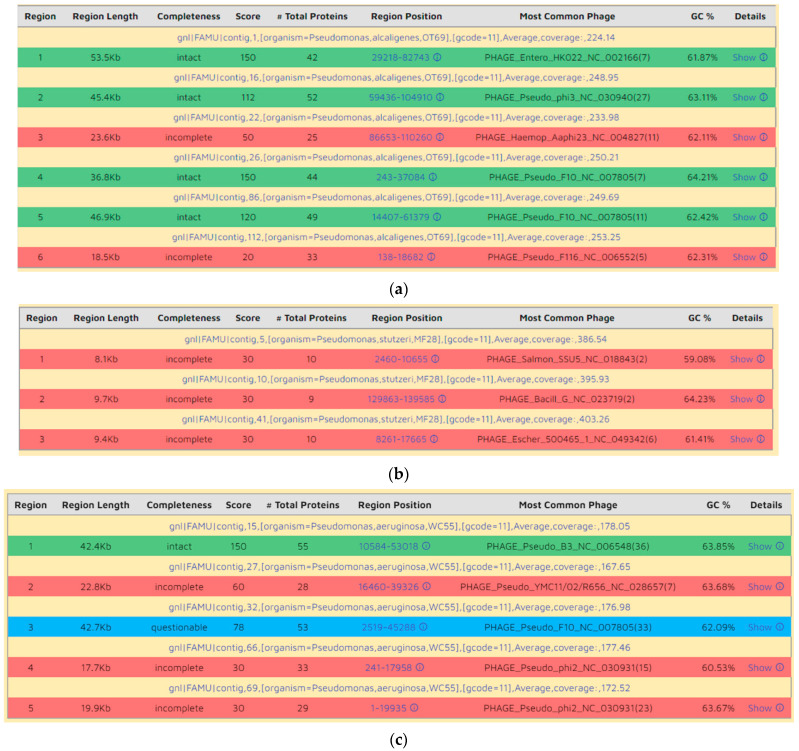
Remnants of bacteriophage genomic regions identified from querying the whole genome sequence of (**A**), *Pseudomonas alcaligenes* OT69; (**B**), *Pseudomonas stutzeri* MF28 and (**C**), *Pseudomonas aeruginosa* WC55. The boxes are color coded with intact prophage shown in green color, incomplete in red color and questionable in blue color, respectively.

**Figure 6 microorganisms-09-00490-f006:**
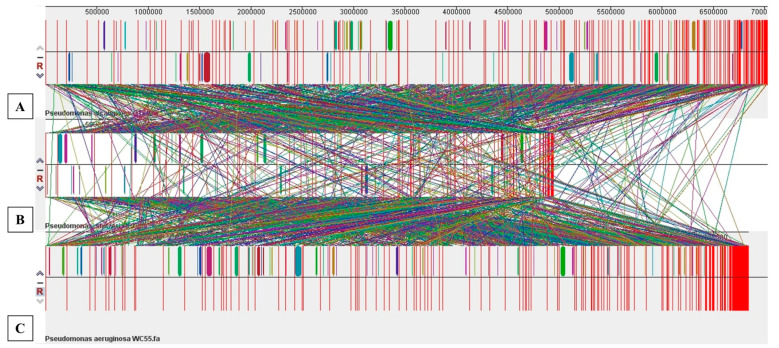
Whole genome comparative alignment of isolated *Pseudomonas* species using MAUVE analysis. Shown are (**A**), *Pseudomonas alcaligenes* OT69; (**B**), *Pseudomonas stutzeri* MF28 and (**C**), *Pseudomonas aeruginosa* WC55, respectively. Each of the genome sequence analyzed is presented horizontally with the scale showing the sequence coordinates and the conserved segments represented as the colored blocks which are connected across genomes. Blocks that are shifted downward in a genome represent those segments that are inverted relative to the reference genomes. The aligned region is in the forward orientation relative to the first genome sequence if a block lies above the center line; blocks below the center line indicate regions that align in the reverse complement (inverse) orientation of the reference genome. Genomic regions falling outside the blocks lack a detectable homology among the genomes analyzed. Within each block is shown the similarity profile of the genome sequence such that the height of the similarity profile corresponds to the level of conservation in that specific region of the genome. White areas represent those genomic regions that did not align well between the input genomes and likely contain sequence elements specific to a particular genome.

**Table 1 microorganisms-09-00490-t001:** Pair-wise comparisons of isolated *Pseudomonas* species with type strains available in the Type (Strain) Genome Server (TYGS) workflow.

*Pseudomonas alcaligenes strain OT69:*								
Query	Subject	*d* _0_	C.I.*d*_0_	*d* _4_	C.I.*d*_4_	*d* _6_	C.I.*d*_6_	Diff. G+C Percent
‘Pseudomonas alcaligenes OT69.fa’	*Pseudomonas otitidis* DSM 17224	52.3	[48.9–55.8]	29.0	[26.6–31.5]	45.4	[42.4–48.4]	0.88
‘Pseudomonas alcaligenes OT69.fa’	*Pseudomonas furukawall* KF 707	30.5	[27.1–34.1]	26.2	[23.9–28.7]	28.4	[25.5–31.5]	0.54
‘Pseudomonas alcaligenes OT69.fa’	*Pseudomonas resinovorans* DSM 21078	31.7	[28.3–35.3]	25.0	[22.7–27.5]	29.1	[26.2–32.2]	2.38
‘Pseudomonas alcaligenes OT69.fa’	*Pseudomonas alcaligenes* NBRC 14159	26.8	[23.5–30.5]	24.4	[22.1–26.9]	25.2	[22.4–28.3]	1.2
‘Pseudomonas alcaligenes OT69.fa’	*Pseudomonas delhiensis* CCM 7361	27.9	[24.5–31.5]	23.9	[21.6–26.3]	25.9	[23.1–29.0]	2.12
‘Pseudomonas alcaligenes OT69.fa’	*Pseudomonas citronellolis* LMG 18378	26.7	[23.3–30.3]	23.8	[21.5–26.3]	25.0	[22.1–28.1]	1.58
‘Pseudomonas alcaligenes OT69.fa’	*Pseudomonas humi* CCA 1	27.1	[23.7–30.7]	23.7	[21.4–26.2]	25.3	[22.4–28.4]	1.25
‘Pseudomonas alcaligenes OT69.fa’	*Pseudomonas guguanensis* JCM 18416	26.1	[22.8–29.7]	23.3	[21.0–25.8]	24.4	[21.6–27.5]	1.75
‘Pseudomonas alcaligenes OT69.fa’	*Pseudomonas nitroreducens* NBRC 12694	26.2	[22.8–29.8]	22.8	[20.6–25.3]	24.4	[21.5–27.5]	0.96
‘Pseudomonas alcaligenes OT69.fa’	*Pseudomonas composti* CCUG 59231	23.9	[20.6–27.6]	22.5	[20.2–24.9]	22.6	[19.8–25.7]	3.55
***Pseudomonas stutzeri strain MF28:***								
**Query**	**Subject**	***d*_0_**	**C.I.*d*_0_**	***d*_4_**	**C.I.*d*_4_**	***d*_6_**	**C.I.*d*_6_**	**Diff. G+C Percent**
‘Pseudomonas stutzeri MF28.fa’	*Pseudomonas xanthomarina* DSM 18231	46.4	[43.1–49.9]	24.9	[22.5–27.3]	39.4	[36.4–42.4]	1.95
‘Pseudomonas stutzeri MF28.fa’	*Pseudomonas nitrititolerans* GL 14	30.9	[27.5–34.5]	23.8	[21.5–26.3]	28.1	[25.2–31.2]	0.76
‘Pseudomonas stutzeri MF28.fa’	*Pseudomonas zhandongensis* NEAUST 5-21	41.9	[38.6–45.4]	23.6	[21.3–26.1]	35.8	[32.8–38.8]	2.7
‘Pseudomonas stutzeri MF28.fa’	*Pseudomonas chloritidismutans* DSM 13592	30.7	[27.3–34.2]	23.0	[20.7–25.4]	27.8	[24.9–30.9]	0.2
‘Pseudomonas stutzeri MF28.fa’	*Pseudomonas kunmingensis* DSM 25974	30.7	[27.3–34.3]	23.0	[20.7–25.5]	27.8	[24.9–30.9]	0.33
‘Pseudomonas stutzeri MF28.fa’	*Pseudomonas nosocomialis* A31/70	26.6	[23.2–30.2]	22.8	[20.5–25.3]	24.7	[21.8–27.8]	3.27
‘Pseudomonas stutzeri MF28.fa’	*Pseudomonas stutzeri* ATCC 17588	31.2	[27.8–34.8]	22.8	[20.5–25.3]	28.1	[25.2–31.2]	1.64
‘Pseudomonas stutzeri MF28.fa’	*Pseudomonas azotifigens* DSM 17556	24.0	[20.7–27.6]	22.7	[20.4–25.2]	22.7	[19.9–25.8]	4.7
‘Pseudomonas stutzeri MF28.fa’	*Pseudomonas songnenensisi* NEAU-ST 5-5T	31.3	[27.9–34.8]	22.6	[20.4–25.1]	28.1	[25.2–31.2]	0.96
‘Pseudomonas stutzeri MF28.fa’	*Pseudomonas balearica* DSM 6083	27.6	[24.4–31.2]	22.3	[20.1–24.8]	25.3	[22.5–28.5]	2.41
***Pseudomonas aeruginosa strain WC55:***								
**Query**	**Subject**	***d*_0_**	**C.I.*d*_0_**	***d*_4_**	**C.I.*d*_4_**	***d*_6_**	**C.I.*d*_6_**	**Diff. G+C Percent**
‘Pseudomonas aeruginosa WC55.fa’	*Pseudomonas aeruginosa* DSM 50071	88.9	[85.5–91.6]	94.7	[93.1–96.0]	92.4	[90.0–94.3]	0.34
‘Pseudomonas aeruginosa WC55.fa’	*Pseudomonas jinjuensis* JCM 21621	27.3	[24.0–31.0]	25.5	[23.2–28.0]	25.8	[22.9–28.9]	0.24
‘Pseudomonas aeruginosa WC55.fa’	*Pseudomonas humi* CCA 1	29.5	[26.1–33.1]	25.3	[23.0–27.8]	27.5	[24.6–30.6]	1.08
‘Pseudomonas aeruginosa WC55.fa’	*Pseudomonas delhiensis* CCM 7361	30.6	[27.2–34.2]	25.3	[23.0–27.8]	28.3	[25.4–31.4]	1.94
‘Pseudomonas aeruginosa WC55.fa’	*Pseudomonas citronellolis* LMG 18378	29.4	[26.0–33.0]	25.2	[22.9–27.8]	27.4	[24.5–30.5]	1.4
‘Pseudomonas aeruginosa WC55.fa’	*Pseudomonas panipatenisis* CCM 7469	28.2	[24.8–31.8]	24.4	[22.1–26.9]	26.2	[23.4–29.4]	0.58
‘Pseudomonas aeruginosa WC55.fa’	*Pseudomonas knackmussii* B 13	27.9	[24.6–31.6]	24.4	[22.1–26.8]	26.1	[23.2–29.2]	0.53
‘Pseudomonas aeruginosa WC55.fa’	*Pseudomonas nitritireducens* WZBFD3-5A2	28.3	[25.0–31.9]	23.8	[21.5–26.2]	26.2	[23.3–29.3]	1.36
‘Pseudomonas aeruginosa WC55.fa’	*Pseudomonas nitroreducens* NBRC 12694	28.6	[25.2–32.2]	23.8	[21.5–26.3]	26.4	[23.5–29.5]	1.13
‘Pseudomonas aeruginosa WC55.fa’	*Pseudomonas japonica* DSM 22348	20.3	[17.2–24.0]	22.0	[19.7–24.4]	19.7	[17.0–22.7]	2.0

## Data Availability

The draft genome sequences of the strains obtained in this study have been deposited as whole-genome shotgun projects in GenBank under the accession numbers ATAQ00000000 (*Pseudomonas aeruginosa* WC55), ATCP00000000 (*Pseudomonas alcaligenes* OT69), ATAR00000000 (*Pseudomonas stutzeri* MF28).

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
