# Peer review of "Comparative Genomic Analysis of Three Pseudomonas Species Isolated from the Eastern Oyster (Crassostrea virginica) Tissues, Mantle Fluid, and the Overlying Estuarine Water Column"

_microorganisms, 2021, doi:10.3390/microorganisms9030490_

Round 1

Reviewer 1 Report

I read with careful attention the manuscript entitled “Comparative Genomic Analysis of Three Pseudomonas species Isolated from the Eastern Oyster (Crassostrea virginica) and it’s Environment” that is an innovative study discussing the environmental importance of oysters as a part of the ecosystem in the USA. In addition to the oyster associated-bacterial genera and their importance in maintaining the ecological functions and vice versa, their role in transmission of different antibiotic resistant gene determinants and the resultant public health hazards. That area, on which a very limited knowledge is available.

The manuscript also discusses the mechanisms of horizontal gene transfer and phage-mediated gene incorporation among Pseudomonas Spp. and development of resistance mechanisms that represents an emerging public health concern. The authors did a valuable work in isolation, sequencing and comparison of different Pseudomonas strains from oysters.

The manuscript is well written: introduction to the manuscript is clear, experimental results are adequately described, their interpretation is reasonable. However, figures and tables are not clearly readable and have to be improved (e.g., font is too small, poor resolution, numbers should be written for instance. Discussion is well organized and presented.

this manuscript will provide helpful information to the readers in the field who are interested in this topic. Overall, I believe that this manuscript is comprehensive and well written and is valid for publication after few modifications.

Author Response

Authors Response: We greatly appreciate your support in reviewing our work which will further strengthen the presentation. Overall, your comments are very  encouraging so thank you for your detailed review.

You do state that the figures and tables are not clearly readable and have to be improved (e.g., font is too small, poor resolution, numbers should be written for instance. Note that The figures and table obtained in this study are directly produced by the workflow or pipelines used and we have no control on their resolutions. 

We have however placed the figures and table where they are first referred to in the manuscript and have made an attempt to name them properly and enhanced their resolutions, where appropriate. Hope this revisions is acceptable to you.

Reviewer 2 Report

The authors present an interesting and comprehensive study on the genomes of three isolates belonging to Pseudomonas genus. Overall, the methods used to study the bacteriophage are appropriate and the results are presented in a clear way, however a few minor issues should be addressed to accept the manuscript for publishing.

Minor issues, comments, suggestions

A rationale of genomic comparisons between the three isolated Pseudomonads (fig. 6) is not fully clear or justified. How the results and conclusion would change if another set of genomes would be chosen for comparison? Are these three isolates of Pseudomonas dominating in the appropriate ecological niches.

Table 1 should be presented as a table but not as an inserted picture.

Figure 2. Fonts should be enlarged. The branches of the trees should be optimized to decrease a void space between them.

Figure 5 should be converted to a table but not be presented as a set of the printed screens.

The References should be presented in the form required.

Reviewer 3 Report

The authors proposed to study three pseudomonas species isolated from the eastern oyster and its environment through comparative genomic analysis. I should appreciate the authors’ efforts to make some results. Followings are some comments to the authors to improve quality of the manuscript. In title, “it’s” should be “its”. The quality of all tables and figures is very poor. The text in tables and figures is unreadable. Please improve the resolution. Please add legends into figures 2-6. 4. Authors are requested to publish the codes and data, which will help relevant researchers. 5. There are many grammar and format mistakes.

Author Response

We greatly appreciate your support in reviewing our work which will further strengthen the presentation. Overall, your comments are very encouraging so thank you for your detailed review.

1. You do have concerns on the title so we have revised the title as follows: Comparative Genomic Analysis of Three Pseudomonas species Isolated from the Eastern Oyster (Crassostrea virginica) Tissues, Mantle Fluid and the Overlaying Estuarine Water Habitat. This title is more specific to the study and better captures the essence of our work.

2. You state that the quality of all tables and figures is very poor. The text in tables and figures is unreadable. Please improve the resolution. 

We have inputted the figures and table in the main document after they were referred to the first time. Note that The figures and table obtained in this study are directly produced by the workflow or pipelines used and we have no control on their resolutions. However, for some figures, we were able to enhance the resolutions, where appropriate. 

3.  Please add legends into figures 2-6. All figures have their legends at the bottom of the figures.

4. Authors are requested to publish the codes and data, which will help relevant researchers. All codes, pipelines and workflows are listed in the manuscript. Methods section also specifies how they were used to generate the figures.

5. There are many grammar and format mistakes. We have combed through the manuscript and addressed any typographical or grammatic mistakes we found.

Thank you for your diligent review and we hope this version will be acceptable for publication!

Round 2

Reviewer 3 Report

The authors have addressed all my comments.